



# Aerosol cloud interaction in the atmospheric chemistry model GRAPES_Meso5.1/CUACE and its impacts on mesoscale numerical weather prediction under haze pollution conditions in Jing-Jin-Ji in China

Wenjie Zhang[1,2], Hong Wang[1], Xiaoye Zhang[1], Liping Huang[3], Yue Peng[1], Zhaodong Liu[1], Xiao Zhang[4], Huizheng Che[1]

[1]State Key Laboratory of Severe Weather & Key Laboratory of Atmospheric Chemistry of CMA, Chinese Academy of Meteorological Sciences, Beijing, China

[2]Department of Atmospheric and Oceanic Sciences & Institute of Atmospheric Sciences, Fudan University, Shanghai, China

[3]Earth System Modeling and Prediction Centre, China Meteorological Administration, Beijing, China

[4]Department of Atmospheric Sciences, Yunnan University, Kunming, China

*Correspondence to*: Hong Wang (wangh@cma.gov.cn)

**Abstract.** The representation of aerosol-cloud interaction (ACI) and its impacts in the current climate or weather model remains a challenge, especially for the severely polluted region with high aerosol concentration, which is even more important and worthy of study. Here, ACI is first implemented in the atmospheric chemistry model GRAPES_Meso5.1/CUACE by allowing for real-time aerosol activation in the Thompson cloud microphysics scheme. Two experiments are conducted focusing on a haze pollution case with coexisted high aerosol and stratus cloud over the Jing-Jin-Ji region in China to investigate the impact of the ACI on the mesoscale numerical weather prediction (NWP). Study results show that the ACI increases cloud droplets number concentration, water mixing ratio, liquid water path (CLWP), and optical thickness (COT), as a result, improving the underestimated CLWP and COT (reducing the mean bias by 21% and 37%, respectively) over a certain subarea by the model without ACI. Cooling in temperature at daytime below 950 hPa occurs due to ACI, which can reduce the mean bias of 2 m temperature at daytime by up to 14% (~0.6 ℃) in the subarea with the greatest change in CLWP and COT. The 24 h cumulative precipitation in this subarea corresponding to moderate rainfall events increases with reduced the mean bias by 18%, depending on the enhanced melting of the snow by more cloud droplets. In other areas or periods with a slight change in CLWP and COT, the impact of the ACI on NWP is not significant, suggesting the inhomogeneity of the ACI. This study demonstrates the critical role of the ACI in the current NWP model over the severely polluted region and the complexity of the ACI effect.



## 1 Introduction

Cloud covers approximately 70% of the Earth's surface, which plays key roles in Earth's radiation budget, hydrologic cycle, and chemical reactions of gaseous and particulate materials (Ramanathan et al., 2005; Rosenfeld et al., 2014). Accurate simulation of cloud in the numerical weather prediction (NWP) model is one of the most important elements in the weather forecast (Seifert et al., 2012; Makar et al., 2015; Wang et al., 2014; Xu et al., 2022).

Aerosol is a key factor for cloud formation—no aerosol, no cloud (Andreae and Rosenfeld, 2008; Pruppacher and Klett, 1980; Mcfiggans et al., 2006). The influence of aerosol on cloud is mainly reflected in two aspects: under the conditions of holding liquid water content constant, more cloud condensation nuclei (CCN) produce smaller but more cloud droplets, causing the albedo of the cloud to be larger (Twomey, 1977); the smaller cloud droplets reduce the collision rate, changing the liquid water content and thickness of the cloud and prolonging of the cloud lifetime (Albrecht, 1989). Aerosol-cloud interaction (ACI) has been the largest uncertainty factor in the climate and weather forecast (Quaas, 2015; Myhre et al., 2013; Makar et al., 2015). One of the key potential challenges is to be defined the ability of aerosol to act as cloud droplets (Chang et al., 2021; Rosenfeld et al., 2019; Che et al., 2017; Sun and Ariya, 2006).

In the current NWP model, the cloud microphysics scheme determines the evolution of hydrometeors (Listowski and Lachlan-Cope, 2017). However, the number concentration of cloud droplets in most cloud microphysics schemes is usually set to be constant in NWP model (i.e., space-time invariant) (Thompson et al., 2008; Thompson et al., 2004; Hong and Lim, 2006; Morrison et al., 2009), which ignores the impact of aerosol on the cloud. Even in schemes that can predict the number concentration of cloud droplets, such as the WDM6 scheme (the initial CCN is a constant) (Lim and Hong, 2010) and Thompson scheme (a preset aerosol emission) (Thompson and Eidhammer, 2014), the impact of aerosol is still not fully considered. According to previous studies, there are huge differences in the anthropogenic aerosol emission globally (e.g., higher aerosol loading over northern India and eastern China) (Che et al., 2015), and the response of cloud physical properties to aerosol is obvious (Miltenberger et al., 2018; Lawand et al., 2022; Mccoy et al., 2018; Zheng et al., 2018b). The lack of anthropogenic aerosol emission, bringing large simulation errors, cannot meet the requirements of weather forecast by the NWP model, especially in precipitation and temperature predictions (Su and Fung, 2018; Zhang et al., 2015; Huang and Ding, 2021). For example, in the Global Forecast System (GFS) model without aerosol feedback, the simulation of 2 m temperature showed larger errors when heavy aerosol pollution or thick cloud cover occurs (Huang and Ding, 2021).

Recognizing the importance of aerosol changes to the cloud, weather, chemistry, etc., many studies have incorporated ACI effects into the NWP models to evaluate the impact of the ACI (Zhao et al., 2017; Zhou et al., 2016; Miltenberger et al., 2018; Wong, 2012; Makar et al., 2015). The study results show that the ACI significantly increases the number concentration of cloud droplets and liquid water content during the selected study period and further leads to a decrease in surface downward short-wave radiation (SDSR), boundary layer height, and surface temperature (Makar et al., 2015; Zhang et al., 2015; Zhang et al., 2010). As a result, the simulated errors in precipitation and temperature are reduced (Zhou et al., 2016). In addition, a recent study using the two-way coupled



Weather Research and Forecasting and Community Multi-scale Air Quality (WRF-CMAQ) model to conduct long-term (2008-2012) simulations in the contiguous US indicates that the main simulated meteorological factors (e.g., temperature, precipitation, wind speed) and air pollutants (e.g., ozone,

sulfate, nitrate) show improved performance compared to the original model (Wang et al., 2021). These studies further prove the critical role of the ACI in the NWP model, yet the ability to consider the ACI effect in weather forecast is still poor. Meanwhile, due to the predominantly extremely inhomogeneous ACI in time and space, especially under haze pollution conditions, the significance of the ACI effect may not be fully realized in long-term or large-scale studies, thus putting its focus on the weather scale

NWP in severe aerosol polluted Jing-Jin-Ji in China is essential and meaningful.

In this paper, the real-time ACI is first coupled into the atmospheric chemistry model GRAPES_Meso5.1/CUACE for the study of the impact of ACI on the cloud, temperature, and precipitation predictions under haze pollution conditions in Jing-Jin-Ji in China. A representative case, the haze pollution episode from 4 to 8 January 2017 with coexisted severe aerosol pollution and stratus

cloud, is selected to be as the research object. Through this short-term case study, the operating mechanism of ACI in the current model and the spatiotemporal inhomogeneous ACI effect under haze pollution conditions can be clearly understood.

## 2 Materials and Methods

### 2.1 Data used

Hourly $PM_{2.5}$ observation data (unit: μg m$^{-3}$) are provided by more than 1300 air pollution stations (Figure 1) from the Chinese Ministry of Ecology and Environment. Hourly surface meteorological observation data come from the automatic weather stations (Figure 1) of the China Meteorological Administration (CMA), including temperature (unit: ℃) and precipitation (unit: mm). Daily aerosol optical depth (AOD), cloud top pressure (CTP, unit: hPa), cloud optical thickness (COT), cloud liquid

water path (CLWP, unit: g m$^{-2}$), and cloud fraction (CF, unit: %) data are from the Suomi National Polar-orbiting Partnership (SNPP) Visible Infrared Imaging Radiometer Suite (VIIRS) NASA Level-3 (L3) aerosol and cloud properties continuity product with a spatial resolution 1° (https://ladsweb.modaps.eosdis.nasa.gov/search/order/1/CLDPROP_D3_VIIRS_SNPP--5111/). The Number concentration of cloud droplets can be derived based on CLWP, COT, and CF from previous

studies (Bennartz, 2007; Pawlowska and Brenguier, 2000). SDSR data (unit: W m$^{-2}$) are derived from the Clouds and the Earth's Radiant Energy System (CERES) project L3 product, which provides satellite-based observations of Earth's radiation budget (ERB) and cloud with a spatial resolution 1° (https://asdc.larc.nasa.gov/data/CERES/SYN1deg-1Hour/Terra-NPP_Edition1A/2017/01/). The vertical profiles of aerosol and cloud data are provided by the Cloud-Aerosol Lidar and Infrared Pathfinder

Satellite Observation (CALIPSO) Level 2 (L2) vertical feature mask (VFM) data product (https://asdc.larc.nasa.gov/data/CALIPSO/LID_L2_VFM-Standard-V4-20/2017/01/). All data ranges are from 4 to 8 January 2017. National Centers for Environmental Prediction (NCEP) Final analysis (FNLs) data with 0.25° horizontal resolution and 6 h interval (https://rda.ucar.edu/datasets/ds083.3/) are used as meteorological boundary conditions and initial fields in the model. The anthropogenic emission



data entered into the model are the Multi-resolution Emission Inventory for China (MEIC) of Tsinghua
University in December 2016, which covers more than 700 anthropogenic emission sources on China's
mainland (Li et al., 2014; Zheng et al., 2018a; Li et al., 2017).

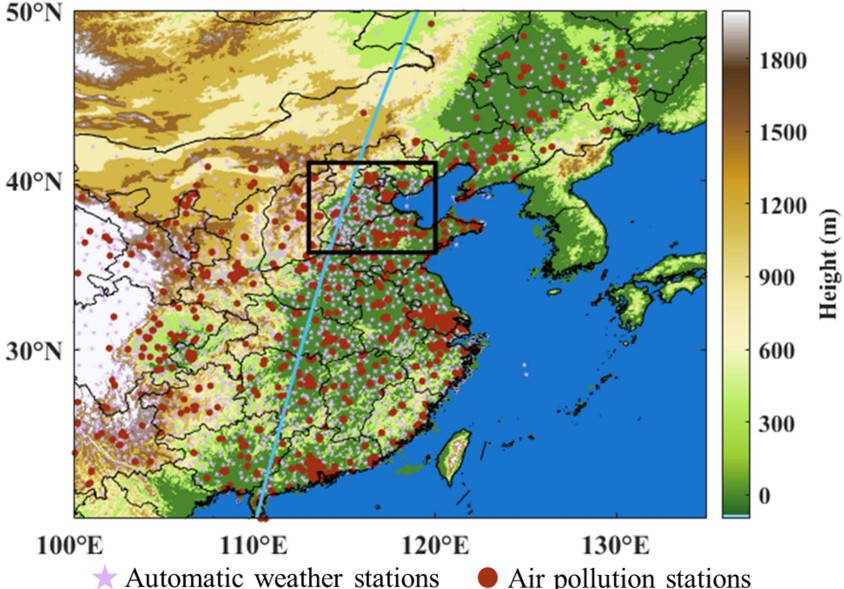

**Figure 1: The map and topographic height of the simulated domain. The turquoise line represents a part of**
**the CALIPSO satellite orbit tracks at 18:12 on 7 January 2017, the black rectangle represents the location of**
**Jing-Jin-Ji, the purple five-pointed stars are the automatic weather stations, and the dark red dots are the**
**air pollution stations.**

### 2.2 Model introduction

The updated operational atmospheric chemistry model GRAPES_Meso5.1/CUACE model mainly
includes four modules: Pre-processing and Quality control, Standard initialization, assimilating
forecasting, and Post-processing (Chen, 2006; Chen et al., 2008; Zhang and Shen, 2008; Wang et al.,
2010), is developed by CMA. The dynamic frame includes an Arakawa C staggered grid, a
semi-implicit and semi-Lagrangian scheme for temporal and advection discretion, and a height-based
terrain-following coordinate. The selected physical-chemistry options include RRTM long-wave
radiation (Mlawer et al., 1997), Thompson cloud microphysics (Thompson et al., 2008), Goddard
short-wave radiation (Chou et al., 1998), Noah land-surface (Chen and Dudhia, 2001), MRF planetary
boundary layer (Hong and Pan, 1996), KFeta cumulus cloud (Kain and Fritsch, 1993), SFCLAY
surface-layer (Pleim, 2007), RADM II gas-phase chemistry (Stockwell et al., 1990), and CUACE
aerosol (Gong and Zhang, 2008; Zhou et al., 2012) schemes. In the RADM II gas-phase chemistry
scheme, 63 gas species through 21 photochemical reactions and 136 gas-phase reactions participate in
the calculations. In the CUACE aerosol scheme (Wang et al., 2010; Gong and Zhang, 2008), 7 types of
aerosol (sea salt, sand/dust, black carbon, organic carbon, sulfate, nitrate, and ammonium salt) are



calculated by hygroscopic growth, dry and wet depositions, condensation, nucleation, etc. These aerosols (except for ammonium salt) are divided into 12 bins with diameter ranges of 0.01-40.96 μm.

The simulated domain of the model covers eastern China (100°E-135°E, 20°N-50°N) (Figure 1) with a horizontal resolution of 0.1°× 0.1° and the 49 vertical layers from the ground (about 52 m) to ~31 km. The study period is from 4 to 8 January 2017 with 72 h prediction time. The spin-up time is 72 h.

### 2.3 Achievement of ACI in the model

To account for the indirect effect of aerosol, we first update the Thompson cloud microphysics scheme
from the original version in the model to the "aerosol-aware" version based on previous studies (Thompson and Eidhammer, 2014; Thompson et al., 2008). The new version includes the activation of water-friendly aerosol to cloud droplets and ice-friendly aerosol to ice crystals. The source of water-friendly aerosol derives from the preset aerosol emission based on the climatological mean state. Second, the assumed aerosol concentration is replaced by real-time simulated aerosol concentration by
CUACE. Water-friendly aerosol number concentration (/kg) required by the activation in the cloud microphysics scheme are calculated by aerosol mass concentration at each grid point according to equations (1), (2), and (3):

$$m_{num} = \frac{4}{3} * \pi * r_{num}^3 * (\rho_{num}) \tag{1},$$

$$N(i, k, j, num) = tracer(i, k, j, num)/m_{num} \tag{2},$$

$$NWFA2(i, k, j) = \sum_{num=1}^{49} N(i, k, j, num) \tag{3}.$$

Here, the m is the aerosol mass, the num is the tracer number from 1 to 49, the r is the mean radius, the ρ is the aerosol density, the tracer is the output aerosol mass concentration, the N is the aerosol number concentration, and the NWFA2 is the total water-friendly aerosol number concentration. I, j, and k represent the grid point. The controversial black carbon and sand/dust in the activation are ignored in
this study. The calculated NWFA2 is input into the cloud microphysics scheme instead of the original assumed aerosol number concentration (Figure 2). Finally, the cloud physical parameters (cloud water and cloud ice effective radius (Rc and Ri)) from the Thompson scheme are input into the Goddard short-wave radiation scheme for radiation calculation and ACI is then completed in the current GRAPES_Meso5.1/CUACE model (Figure 2).

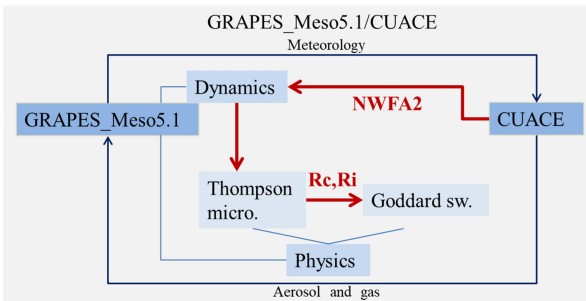


**Figure 2: Diagram of Aerosol cloud interaction in GRAPES_Meso5.1/CUACE model.**

### 2.4 The Case Description



This case is a haze pollution episode (from 4 to 8 January 2017) over Jing-Jin-Ji accompanied by the appearance of stratus cloud, which demonstrates the rationality and pertinence of this simulated study. During this episode, the peak mass concentration of $PM_{2.5}$ exceeds 200 μg m$^{-3}$ (i.e., heavy aerosol pollution occurs in Jing-Jin-Ji) (Figure S1(a)). Besides, moderate rainfall events (10 mm<24 h cumulative precipitation<25 mm) occur in the southeast of Jing-Jin-Ji on 7 January 2017. Light rainfall events (24 h cumulative precipitation<10 mm, including drizzle) contribute to the amount of precipitation in Jing-Jin-Ji on other days. Figure S2(b) shows the vertical distribution of aerosol and cloud layers in Jing-Jin-Ji at 18:12 on 7 January 2017. The aerosol layer is partly overlapped with the cloud layer, suggesting the potentiality of aerosol as CCN and ACI in this region. Besides, different types of cloud can be identified by CTP and COT from satellite data, including stratus, cumulus, cirrus, etc., according to the International Satellite Cloud Climatology Project (ISCCP) (Rossow and Schiffer, 1991; Hahn et al., 2001). Figure S2(a) shows the daily mean CTP and COT in Jing-Jin-Ji from 4 to 8 January 2017. It can be confirmed that the types of cloud over Jing-Jin-Ji are basically stratus cloud (except for 4 January 2017) with lower cloud base height.

### 2.5 Experimental Design

To investigate the ACI and its feedback on the simulated cloud, temperature, and precipitation in the current model, we conduct two experiments (E1 and E2) as shown in Table 1. The E1 experiment is the control experiment with the constant 100 cm$^{-3}$ number concentration of cloud droplets, which is the default setting in the Thompson cloud microphysics scheme. The E2 experiment includes the ACI combined with real-time aerosol activation. The difference in simulations between the E2 and E1 experiment can be attributed to the impact of ACI on current NWP predictions.

**Table 1: The setup of two experiments in the model.**

| Experiment | Description |
| --- | --- |
| E1 | Model run without ACI |
| E2 | Model run with ACI |

### 3 Results

#### 3.1 Model Evaluation

Figure 3(a) and (b) show the spatial distributions of observed and simulated (E1 experiment) mean $PM_{2.5}$ mass concentration during the whole study period, both of which indicate that there is a high-value center of $PM_{2.5}$ mass concentration in the southwest of Jing-Jin-Ji. However, the simulations in the southeast of Jing-Jin-Ji are lower than the observations. The model also captures the observed temporal variation of $PM_{2.5}$ mass concentration, including the rising and falling period, and the correlation coefficient (R) is 0.75 (Figure. S1(a)). To further evaluate the aerosol information at the boundary layer height, Figure S1(b) and (c) show the spatial distributions of the time-average AOD from the E1 experiment and VIIRS. The simulated AOD is consistent with the VIIRS, both exhibiting a



high-value center of AOD similar to $PM_{2.5}$ mass concentration. All these results indicate that the model can accurately reproduce the aerosol pollution level reasonably in Jing-Jin-Ji.

In addition to ensuring reasonably aerosol simulations, it is necessary to have a brief understanding of simulated performance in meteorological factors from the current model without the ACI (the E1 experiment). As shown in Figure 3(c–j), the model basically reproduces the location of the large-scale stratus cloud and accurately simulates the distributions and magnitudes of mean 2 m temperature at daytime (i.e., from the 08:00 to 16:00 local time) and 24 h cumulative precipitation in China. However, compared with VIIRS, the simulated mean COT and CLWP in Jing-Jin-Ji for 5 days (JJJ-5d) show

obvious negative bias (i.e., bias=$X_{sim}$-$X_{obs}$ where $X_{sim}$ and $X_{obs}$ represent the simulations and observations) (-18.4 and -104.2 g m$^{-2}$). Besides, the mean bias of the 2 m temperature at daytime and 24 h cumulative precipitation for JJJ-5d are 3.2 ℃ and -0.11 mm against observations. It can be seen that 2m temperature at daytime is overestimated and 24 h cumulative precipitation is underestimated by the E1 experiment (GRAPES_Meso5.1/CUACE model without ACI), especially in the southern part of

Jing-Jin-Ji with more cloud cover.

**Figure 3: Comparisons of observed and simulated mean (a) and (b) PM$_{2.5}$ mass concentration, (c) and (d)**



COT, (e) and (f) CLWP, (g) and (h) 2 m temperature at daytime, and (i) and (j) 24 h cumulative precipitation from 4 to 8 January 2017. The black rectangle represents the location of Jing-Jin-Ji.

### 3.2 The impact of ACI on cloud

When the ACI is activated in the model (i.e. the E2 experiment), there are more generated cloud droplets and more reasonable distributions of cloud droplets (Figure S3), compared with the constant number concentration of cloud droplets (100 cm$^{-3}$) in the E1 experiment. Furthermore, the Rc decreases (Figure omitted) due to competitive growth. Such changes have impacts on hydrometeors in the cloud (Lohmann and Feichter, 2005). Figure 4 shows the temporal variation of the regional mean hydrometeors mixing ratio in Jing-Jin-Ji from the E1 and E2 experiments. On the whole, the cloud top height is above the 0 °C isotherm and the magnitude of snow mixing ratio (Qs) is relatively larger, indicating that this cloud system is the mixed-phase cloud with more significant cold cloud processes. Taking a day (7 January 2017) as an example (Figure 4 and S4), compared with the E1 experiment, we find that the cloud water mixing ratio (Qc) increases significantly (the maximum increase in the vertical direction is more than $4\times10^{-3}$ g kg$^{-1}$) in the E2 experiment. This is mainly due to the smaller Rc and lower auto-conversion of cloud water to form rain. Typically, during the warm cloud process, inhibited auto-conversion of cloud water may reduce the rainwater mixing ratio (Qr). However, Qr increases in the E2 experiment, which is mainly related to the increased rain water from enhanced snow melting. This phenomenon also reflects the characteristics of the cold cloud processes. To further confirm the changes in snow, we find a significant increase of Qs in the mid-troposphere, which promotes the melting of the snow to form rain. The increase of Qs in the E2 experiment is mainly because the ACI increases the supercooled cloud water in the mid-troposphere and may promote the riming growth process. The Qs, in addition, decreases in the lower troposphere, which may be related to the melting of snow to form rain. The changes in ice mixing ratio (Qi) and graupel mixing ratio (Qg) are relatively small. It should be noted that, in the E2 experiment, additional new cloud do not generate in the original area without cloud (the E1 experiment), even though the ACI is activated. For example, on 5 January 2017 (Figure 4), the original model do not reproduce the fact that VIIRS indicated the presence of cloud in Jing-Jin-Ji and the ACI effect also do not improve this phenomenon, indicating the limitations of the ACI. More detailed studies are needed in the future.



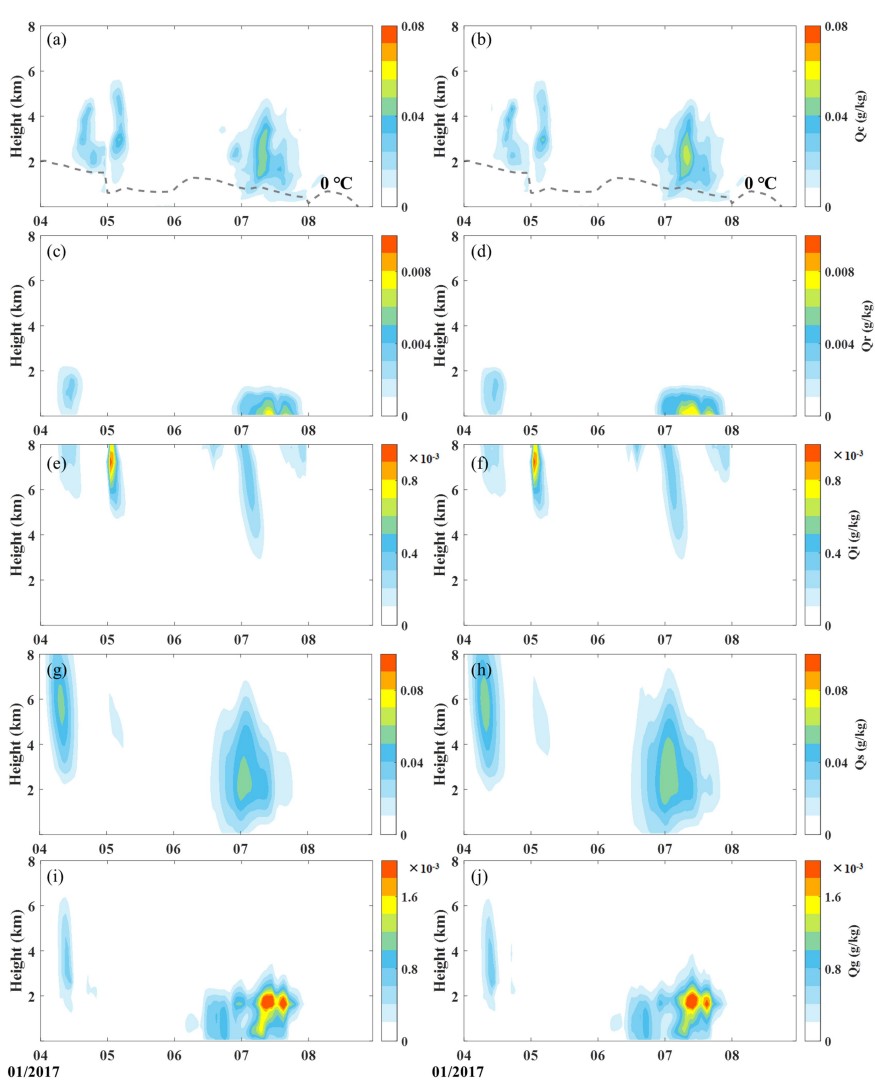

**Figure 4: The vertical distribution of regional mean hydrometeors mixing ratio (Qc, Qr, Qi, Qs, and Qg) in Jing-Jin-Ji from the (the left column) E1 and (the right column) E2 experiment.**


The cloud macroscopic characteristics can be affected accordingly. The ACI increases simulated CLWP and COT, both of which are more consistent with satellite observations with slightly reduced the mean bias for JJJ-5d by 2% and 2%, respectively (Figure 10(a) and (b)). In particular, on 7 January 2017, the

daily mean CLWP and COT increase significantly in most areas of Jing-Jin-Ji due to the ACI (Figure 5 and 6). The maximum values of increase are 137.7 g m$^{-2}$ and 25.1, respectively. This increase reduces the regional mean bias of the E1 experiment by 7% (from -163.4 to -151.8 g m$^{-2}$) for CLWP and 7% (from -22.3 to -20.7) for COT against the VIIRS. In addition, it can be seen that the impact of the ACI





on CLWP and COT are significantly different in various regions of Jing-Jin-Ji. We explain this

phenomenon in Section 3.4.

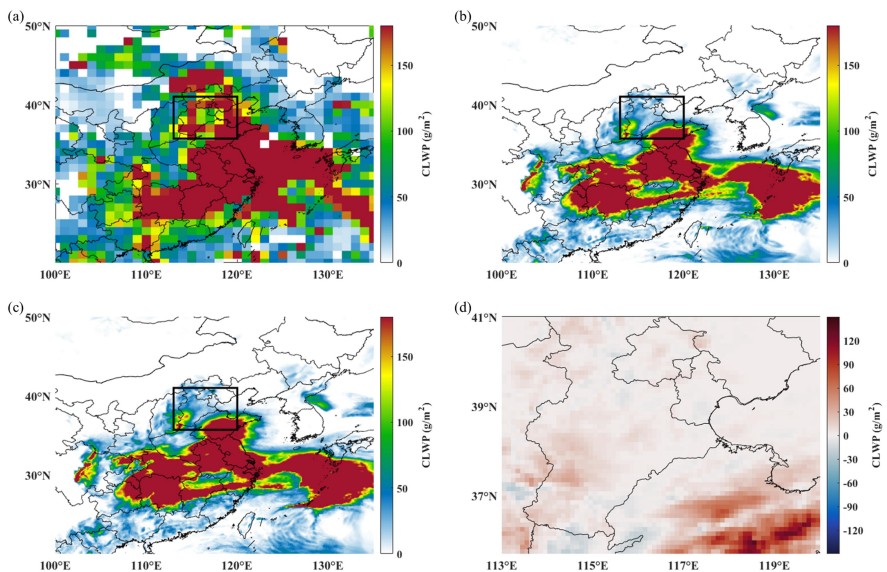

**Figure 5: The spatial distribution of mean CLWP on 7 January 2017. (a) The VIIRS. (b) The E1 experiment. (c) The E2 experiment. (d) The difference between the E2 and E1 experiment in Jing-Jin-Ji.**

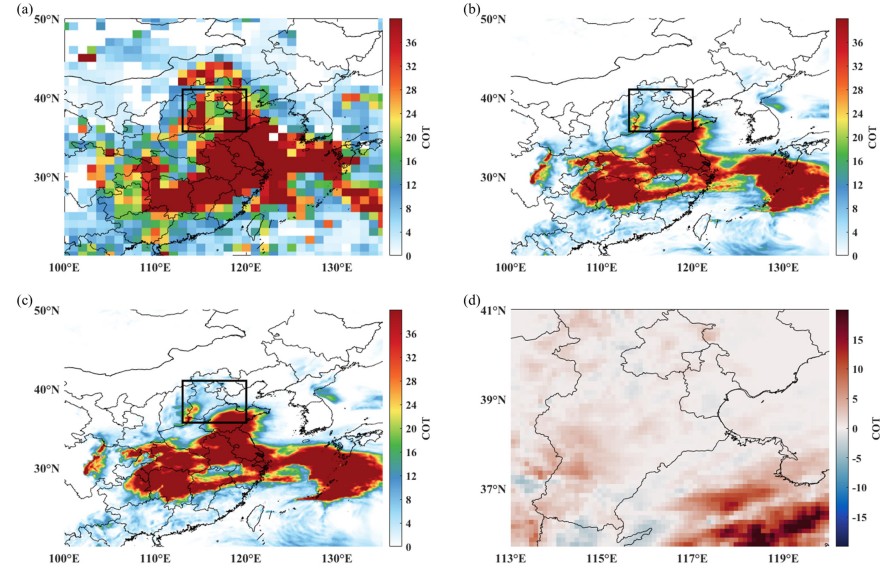


**Figure 6: As in Figure 5, but for mean COT on 7 January 2017.**



### 3.3 The impact of ACI on NWP

Changes in simulated cloud inevitably affect NWP such as radiation, temperature, precipitation, etc. (Liu et al., 2019; Borys et al., 2000). The above section shows that the ACI effect significantly influences the cloud's micro and macro physical properties. Furthermore, the spatial distribution of daily mean SDSR on 7 January 2017 is shown in Figure S5. Compared with the E1 experiment, the decreased SDSR from the E2 experiment has better performance against the data from CERES, with

the regional mean bias of 44.6 vs. 42.9 W m$^{-2}$ in Jing-Jin-Ji. The maximum value of decrease can reach up to 26.1 W m$^{-2}$. Evaluations in Section 3.1 have pointed out that the simulated 2 m temperature at daytime from the E1 experiment has a significantly positive bias in Jing-Jin-Ji, further supported by simulations on 5 and 7 January 2017 (Figure 7(b) and (e)). The simulated mean 2 m temperature at daytime by the E2 experiment with ACI has significantly decreased in cloudy fields of Jing-Jin-Ji with

the maximum decrease value of 1 ℃ on 7 January 2017 (Figure 7(f)). However, other days during the study period are not significantly affected. For example, on 5 January 2017, the highest absolute difference of 2 m temperature at daytime between the E2 and E1 experiment is less than 0.2 ℃ (Figure 7(c)). In summary, the regional mean bias of 2 m temperature at daytime has been slightly improved (2%) for JJJ-5d (3.2 ℃ for the E1 experiment vs. 3.1 ℃ for the E2 experiment) (Figure 10(c)); while

this improvement on 7 January 2017 increases to 4% with the bias of 2.7 vs. 2.6 ℃. Figure 8 shows the difference in temperature at daytime in the vertical direction between the E2 and E1 experiment on 7 January 2017. The decrease in temperature at 1000 and 950 hPa (Figure 8(a) and (b)) is more significant than those at 900 and 850 hPa (Figure 8(c) and (d)). The maximum value of decrease at 1000 hPa is more than 0.8 ℃. As for the temperature above 700 hPa, the changes in temperature are

not significant with the maximum absolute difference being less than 0.2 ℃ (Figure 8(e) and (f)). This phenomenon suggests that real-time ice nucleation is expected in the following study. Similar to the ACI effect on cloud characteristics, the impact on temperature is inhomogeneous in Jing-Jin-Ji, especially in the lower atmosphere. It is worth noting that the changes in CLWP, COT, and temperature at daytime are all more significant in the same areas or periods, which is emerging evidence for

explaining the inhomogeneous ACI effect.

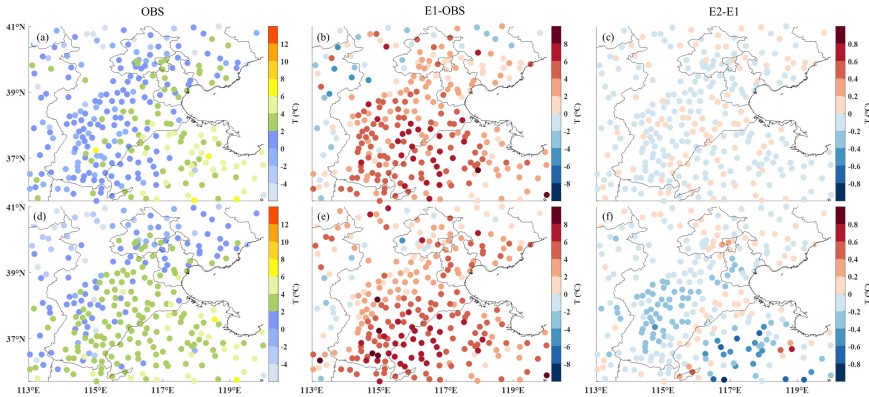

**Figure 7: The spatial distribution of mean 2 m temperature at daytime. (a) and (d) The mean observations.**





**(b) and (e) The bias of the E1 experiment. (c) and (f) The difference between the E2 and E1 experiment. The**

**above row and following row are data on 5 and 7 January 2017, respectively.**

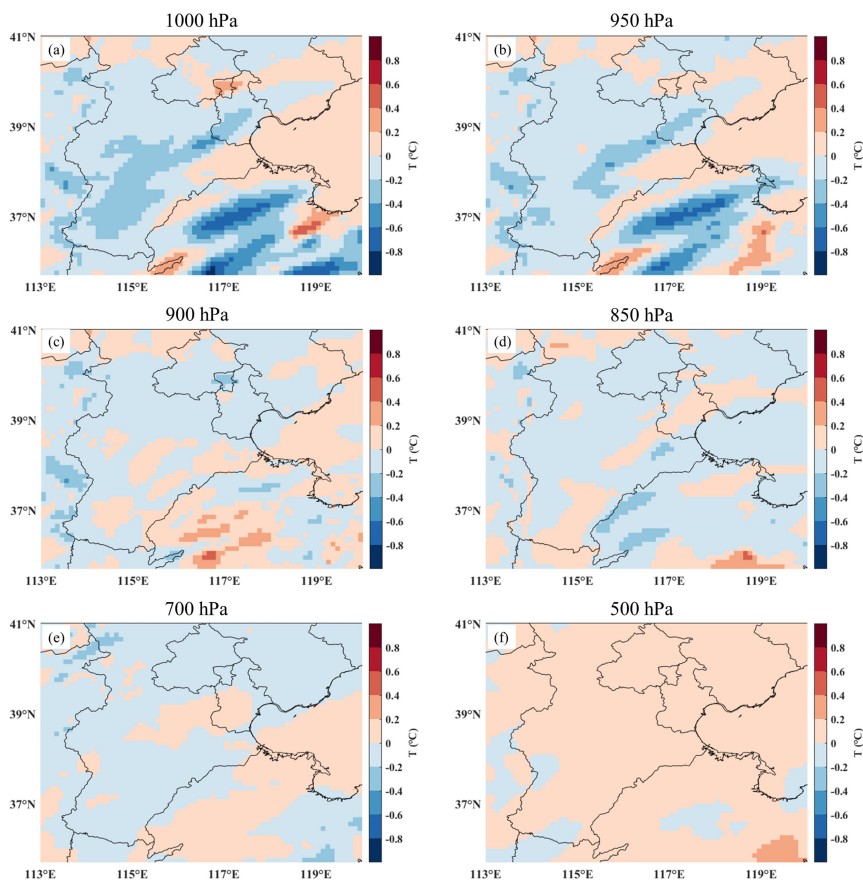

**Figure 8: The difference of mean temperature at daytime on 7 January 2017 between the E2 and E1 experiment in vertical direction. (a) 1000 hPa. (b) 950 hPa. (c) 900 hPa. (d) 850 hPa. (e) 700 hPa. (f) 500 hPa.**


The simulated precipitation is another meteorological factor that requires special attention. Unlike temperature, the impact of ACI on precipitation is more complex. As mentioned in Section 2.4, moderate rainfall events occur on 7 January 2017 in the southeast of Jing-Jin-Ji. A series of stations

with moderate precipitation events can be collectively referred to as the moderate rainfall area (the red oval in Figure 9). Apart from this, other areas in Jing-Jin-Ji are light rainfall areas during the study period. In the moderate rainfall area associated with significant changes in CLWP and COT, the ACI increases 24 h cumulative precipitation with the maximum value exceeding 4.2 mm (Figure 9(d)), which improves underestimated mean precipitation by 26% (Figure 9(c)). Besides, the ACI decreases

24 h cumulative precipitation in light rainfall areas. For example, in a light rainfall area (the black oval

in Figure 9), this decrease due to ACI is observed with the maximum value exceeding -1 mm (Figure 9(d)). In terms of the study period average, the ACI reduces the mean bias of 24 h cumulative precipitation by 7% in these light rainfall areas. The combined effect of ACI on moderate rainfall and light rainfall improves the simulated mean 24 h cumulative precipitation for JJJ-5d with the mean bias

of -0.11 vs. -0.07 mm (Figure 10(d)). The regionality of the ACI effect on precipitation is reflected. In Section 3.4, we continue to quantify the improvement in the selected areas and explore the possible reasons for discrepancies. More detailed evaluations about precipitation will be carried out in future works.

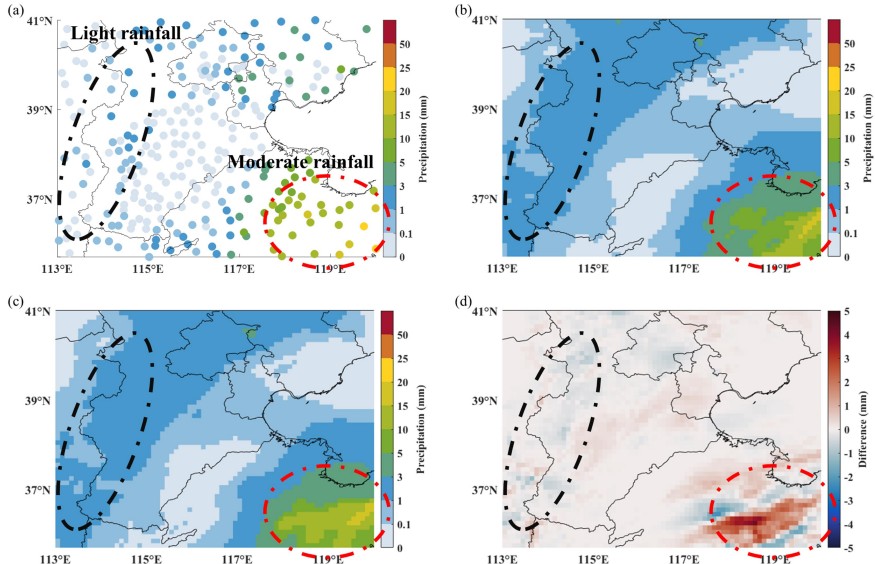

**Figure 9: The spatial distribution of 24 h cumulative precipitation on 7 January 2017. (a) The observations. (b) The E1 experiment. (c) The E2 experiment. (d) The difference between the E2 and E1 experiment. The red and black ovals represent the moderate and light rainfall areas, respectively.**

### 3.4 The variations of ACI effect in different subareas and possible reasons

In Sections 3.2 and 3.3, we have preliminary findings that the ACI effect varies in different areas of
Jing-Jin-Ji. According to the aerosol pollution levels and the magnitude of CLWP on 7 January 2017, domain-A (i.e., DA (113°E-115°E, 36.4°N-38.5°N)) and domain-B (i.e., DB (116.2°E-120°E, 35.7°N-37.5°N)) are selected for key analysis (Figure S6(a) and (b)). In the DA, the ACI increases CLWP and COT with the mean bias decreased by 27% (from -33.8 to -24.5 g m$^{-2}$) and 12% (from -13.8 to -12.1); while in the DB, the more significant increase occurs in CLWP and COT which also helps to
reduce the mean bias by 21% (from -203.2 to -160.1 g m$^{-2}$) and 37% (from -18.7 to -11.7) (Figure 10(a) and (b)). The impact of ACI on cloud is more significant in the DB, which is also the subarea with the greatest change in CLWP and COT in Jing-Jin-Ji. Based on previous studies (Pawlowska and Brenguier, 2000; Lohmann and Feichter, 2005; Zhou et al., 2020; Lu et al., 2012), we explore some possible reasons including aerosol levels, local meteorological conditions, cloud types, etc. The


regional mean PM$_{2.5}$ mass concentration in the DA (164.3 µg m$^{-3}$) is much greater than that in the DB (74.5 µg m$^{-3}$) (Figure S6(a)), suggesting that the ACI effect is not entirely determined by aerosol levels. Supersaturation (900 hPa) and ascent speed (900 hPa), two typical meteorological factors, are shown in Figure S6(c) and (d). Compared with DA, the positive supersaturation and ascent speed control more areas in the DB, which is conducive to cloud droplets nucleation and cloud evolution. More importantly,

the pre-simulated CLWP by the model without ACI in the DB (196.9 g m$^{-2}$) is higher than that in the DA (80.7 g m$^{-2}$), which is also consistent with the variations of the ACI effect. The CLWP provides partial information about meteorological conditions and cloud types, thus we speculate that the inhomogeneous ACI effect on cloud under haze pollution conditions in Jing-Jin-Ji is related to the magnitude of pre-simulated CLWP, which needs further work to verify. In addition, the mean bias 2 m

temperature at daytime is reduced by 10% (from 1.9 to 1.7 ℃) in the DA and 14% (from 4.1 to 3.5 ℃) in the DB (Figure 10(c)), indicating the more significant ACI effect in 2 m temperature at daytime occurs in the subarea with greater change in CLWP and COT. This can also be further proved by comparing the improved 2 m temperature at daytime in four cases (JJJ-5d, JJJ, DA, and DB) with different changes in CLWP and COT (Figure 10(c)). As for the simulated precipitation, the impact of

ACI is related to the response of cold cloud processes to increased cloud droplets. In the DB with moderate rainfall events and the greatest change in CLWP and COT, the ACI increases 24 h cumulative precipitation with the mean bias reduced by 18% (from -2.36 to -1.94 mm) (Figure 10(d)); while in the DA with light rainfall events, the ACI decreases 24 h cumulative precipitation with the mean bias reduced by 3% (from 1.14 to 1.11 mm). According to Section 3.2 and Figure S7, we have enough

evidence to believe that the increased precipitation in the DB is caused by the enhanced melting of the snow to form rain in cold cloud processes. Meanwhile, the decreased precipitation in the DA is associated with inhibited the melting of the snow to form rain due to more and smaller cloud droplets. The less efficient collision and coalescence processes in light rainfall cannot be ignored (Qian et al., 2009). More detailed studies are needed.

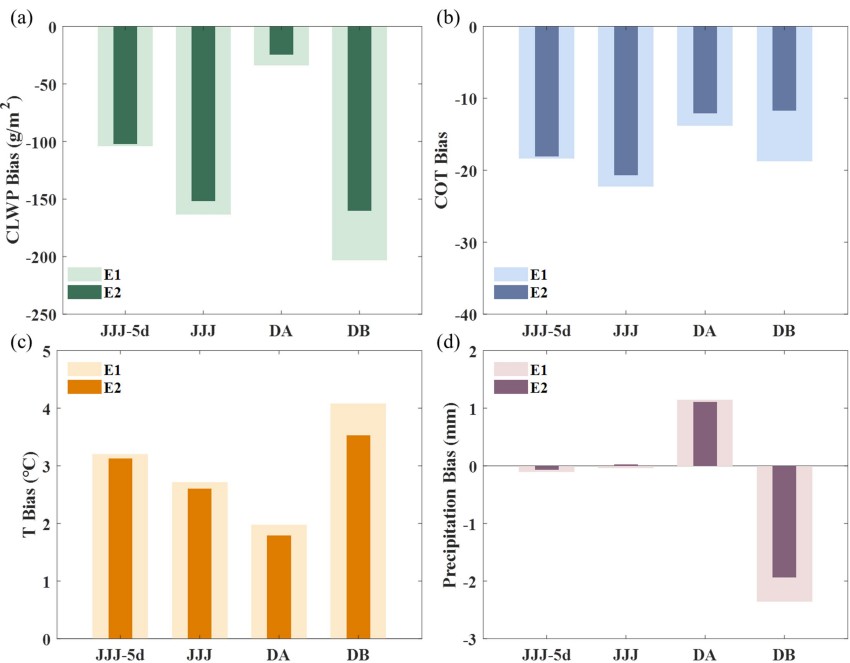

**Figure 10: Comparisons of the mean bias of simulations from the E1 and E2 experiment for 5 days in Jing-Jin-Ji (JJJ-5d) and a day (7 January 2017) in Jing-Jin-Ji (JJJ), DA, and DB. (a) CLWP. (b) COT. (c) 2 m temperature at daytime. (d) 24 h cumulative precipitation.**

## 4 Conclusions

In this work, the ACI is first completed in the GRAPES_Meso5.1/CUACE model by linking the real-time calculated aerosol in CUACE to cloud droplets nucleation in the Thompson cloud microphysics scheme and transferring diagnostic variables (Rc and Ri) to Goddard short-wave radiation scheme. Using this developed model, two experiments, including a control experiment without ACI and a comparative experiment with activated ACI, are conducted to investigate the impact of ACI on simulations (e.g., cloud, temperature, and precipitation) in a typical haze pollution episode (from 4 to 8 January 2017) with heavy aerosol concentration and stratus cloud over Jing-Jin-Ji in China.

The results show that the ACI increases the number concentration of cloud droplets, Qc, CLWP, and COT; and decreases Rc. The increased CLWP and COT are more consistent with satellite observations,



especially in a certain subarea with the mean biases decreased by up to 21% (from -203.2 to -160.1 g m$^{-2}$) and 37% (from -18.7 to -11.7). The cloud extinction enhanced by the ACI accompanied by the decreased SDSR further cools temperature at daytime below 950 hPa, as a result, reducing the regional

mean biases of 2 m temperature at daytime by up to 14% (from 4.1 to 3.5 ℃) in the subarea with the greatest change in CLWP and COT. The 24 h cumulative precipitation in this subarea, corresponding to moderate rainfall events, increases due to the ACI with reduced mean biases by 18% (from -2.36 to -1.94 mm), which is caused by the enhanced melting of the snow to form rain in cold cloud processes. However, in other areas or periods with a slight change in CLWP and COT, the improvement of ACI on

NWP is not significant, suggesting the spatiotemporal inhomogeneous ACI effect.

In general, the GRAPES_Meso5.1/CUACE model coupled with the ACI has a better performance on simulated cloud, temperature, and precipitation under haze pollution conditions in Jing-Jin-Ji. However, the inhomogeneous ACI effect in time and space still needs more detailed work in the future. In addition, there are still some shortcomings worth improving such as aerosol activation in the

convective cloud (Ekman et al., 2011), real-time ice-friendly aerosol input (Demott et al., 2010; Thompson and Eidhammer, 2014), etc.

**Competing Interests**

The authors declare that they have no conflict of interest.

**Author contributions**

Conceptualization: Hong Wang and Xiaoye Zhang. Investigation and Writing: Wenjie Zhang. Methodology: Wenjie Zhang and Liping Huang. Data curation: Yue Peng, Zhaodong Liu, and Xiao Zhang. Supervision: Hong Wang, Xiaoye Zhang, and Huizheng Che.

**Acknowledgment**

This study is supported by the NSFC Major Project (42090030); the National Key Research and
Development Program of China (2019YFC0214601); the NSFC for Distinguished Young Scholars (41825011).

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
