# Peer review of "Aerosol cloud interaction in the atmospheric chemistry model GRAPES\_Meso5.1/CUACE and its impacts on mesoscale numerical weather prediction under haze pollution conditions in Jing-Jin-Ji in China"

_Atmospheric Chemistry and Physics, 2022_

## Author Comment (AC1)

Dear Reviewer:

Thank you for careful comments. These comments are all valuable and very helpful for revising and improving our paper, as well as the important guiding significance to our researches. We have studied the comments carefully and have made corrections which we hope meet with approval. The Reviewer's comments are in blue and our responses are in black. Revised portion are marked in red in the marked-up manuscript. The main corrections in the paper and the point-to-point responses are as following:

Response to Reviewer #1:

Main comments:

**1. The cloud droplets nucleation is a bridge connecting aerosol and cloud droplets and this is also the key of this paper. I think how cloud droplets nucleation is parameterized should be added to the manuscript or supporting material in detail.**

Response: Accept. Thanks for your suggestions. In the new Thompson cloud microphysics scheme, when the supersaturation degree is greater than 0, the water-friendly aerosol can be activated as cloud droplets by equation (1):

$$\text{Activ\_Nc} = \text{NWFA} \times \text{AF} \tag{1},$$

where Activ_Nc represents activated cloud droplets, NWFA represents the number concentration of water-friendly aerosol, and AF represents activated fraction. The activation fraction is determined by the simulated temperature, vertical velocity, number concentration of water-friendly aerosol, and pre-determined values of the hygroscopicity parameter (0.4) and aerosol mean radius (0.04 μm) by using a lookup table. This lookup table is created by the explicit treatment of Köhler activation theory using different number concentration of water-friendly aerosol (10.0, 31.6, 100.0, 316.0, 1000.0, 3160.0, 10000.0 cm$^{-3}$), vertical velocity (0.01, 0.0316, 0.1, 0.316, 1.0, 3.16, 10.0, 31.6, 100.0 m/s), temperature (243.15, 253.15, 263.15, 273.15, 283.15, 293.15, 303.15 K), aerosol mean radius (0.01, 0.02, 0.04, 0.08, 0.16 μm), and

hygroscopicity parameter (0.2, 0.4, 0.6, 0.8) according to previous studies (Feingold, and Heymsfield, 1992; Thompson and Eidhammer, 2014). We have added these introductions of cloud droplets nucleation to the revised manuscript in Text S1.2 in supplementary material.

**References**

Thompson, G. and Eidhammer, T.: A Study of Aerosol Impacts on Clouds and Precipitation Development in a Large Winter Cyclone, Journal of the Atmospheric Sciences, 71, 3636-3658, https://doi.org/10.1175/JAS-D-13-0305.1, 2014.

Feingold, G., and Heymsfield, A. J.: Parameterizations of condensational growth of droplets for use in general circulation models. J. Atmos. Sci., 49, 2325–2342, https://doi.org/10.1175/1520-0469(1992)049,2325:POCGOD.2.0.CO;2, 1992.

**2. Is the increase or decrease in precipitation only due to snow melting?**

Response: Revised. In this study, we find that the ACI decreases the 24 h cumulative precipitation in the DA region and increases the 24 h cumulative precipitation in the DB region on 7 January 2017. As you mentioned, the changes in precipitation are not only dependent on the snow melting. Other source/sink processes (e.g., the rain collecting cloud water, the autoconversion of cloud water to from rain, and the rain collecting snow) of rainwater cannot be ignored. However, based on the analysis of the source/sink of rainwater in the cloud microphysical scheme, we think that the increase or decrease in precipitation due to the ACI is mainly caused by the melting of snow to form rain (i.e., prr_sml). The contributions of other source/sink processes of rainwater are relatively small. The followings are more detailed descriptions.

In the Thompson cloud microphysics scheme, the source/sink of rainwater is calculated by the following equation (2):

Rain tendency= prr_wau + prr_rcw + prr_sml + prr_gml + prr_rcs + prr_rcg - prg_rfz - pri_rfz - prr_rci        (2),

where prr_wau is the autoconversion of cloud water to form rain, prr_rcw is the rain collecting cloud water, prr_sml is the melting of snow to form rain, prr_gml the is melting of graupel to form rain, prr_rcs is the rain collecting snow, prr_rcg is the rain collecting graupel, prg_rfz is the freezing of rainwater into graupel, pri_rfz is the freezing of rainwater into ice, and prr_rci is the rain collecting ice. All of these

processes lead to changes in rainwater.

Figure S7 shows the difference in mean hydrometeors mixing ratio and rain tendency processes between the E2 (ACI) and E1 (NO-ACI) experiment. In the DA, the largest contribution to the decrease in precipitation is the prr_sml, followed by the prr_rcs and other processes (Figure S7 (c)). In the DB, the largest contribution to the increase in precipitation is also the prr_sml, followed by the prr_rcw and other processes (Figure S7 (d)). In summary, the increase or decrease in simulated precipitation due to the ACI is mainly caused by the melting of snow to form rain (i.e., prr_sml), followed by other source/sink processes of rainwater. We have made corresponding corrections for accurate descriptions in Text S2 in supplementary material.

[Figure]

**Figure S7: The difference of mean hydrometeors mixing ratio (top) and rain tendency processes (bottom)**

between the E2 and E1 experiment on 7 January 2017 in the DA (a and c) and DB (b and d).

Minor Comments:

**1. The weather stations in Figure 1 are not marked clearly.**

Response: Revised. We have redrawn the Figure 1 to make the representation of the weather stations clearer. The new Figure 1 is added to the manuscript.

[Figure]

Figure 1: The map and topographic height of the simulated domain. The turquoise line represents a part of the CALIPSO satellite orbit tracks at 18:12 on 7 January 2017, the black rectangle represents the location of Jing-Jin-Ji, the gray cross signs are the automatic weather stations, and the dark red dots are the air pollution stations.

**2. The fonts of some units need to be unified, such as the unit in line 289.**

Response: Accept. The font of unit (℃) in line 292 is corrected as Times New Roman. Besides, we keep the fonts of the units throughout the manuscript unified.

**3. In line 104, 'Number' >> 'number'.**

Response: Accept. The "Number" is corrected as "number" in line 104.

**4. Some abbreviations and symbols should be checked throughout this manuscript.**

Response: Accept. We have rechecked the abbreviations and symbols throughout the manuscript, and made corresponding corrections.

**5. The sentence in lines 124-127 need to be reorganized.**

Response: Accept. The sentence "The updated operational atmospheric chemistry model GRAPES_Meso5.1/CUACE model mainly includes four modules: Pre-processing and Quality control, Standard initialization, assimilating forecasting, and Post-processing, is developed by CMA." has been rewritten as "The updated operational atmospheric chemistry model GRAPES_Meso5.1/CUACE developed by CMA mainly includes four modules: Pre-processing and Quality control, Standard initialization, assimilating forecasting, and Post-processing." in lines 125-127.

**6. I think 'in NWP model' in line 55 is duplicated.**

Response: Revised. We delete "in NWP model" in the original manuscript.

**7. In line 241, 'additional new cloud' is not accurate and leads to a misunderstanding. Maybe change this to 'additional cloud field' or 'additional cloud'.**

Response: Revised. As you mentioned, the "additional new cloud" may lead to a misunderstanding. We have corrected "additional new cloud" to "additional cloud fields" in line 245 in the manuscript.

---

## Author Comment (AC2)

Dear Reviewer:

Thank you for careful comments. These comments are all valuable and very helpful for revising and improving our paper, as well as the important guiding significance to our researches. We have studied the comments carefully and have made corrections which we hope meet with approval. The Reviewer's comments are in blue and our responses are in black. Revised portion are marked in red in the marked-up manuscript. The main corrections in the paper and the point-to-point responses are as following:

Response to Reviewer #2:

Major comments:

**1. Some important details about the model is missing. Please see further comments below.**

Response: Revised. We have added some important details to this paper based on your suggestions, including the cloud droplet activation, calculation of water-friendly aerosol, introduction of supersaturation degree, etc. in Text S1 in supplementary material.

**2. The results are sometimes speculative, especially those in Section 3.4. Besides, in Section 3.4, two sub-regions are selected for analysis; while in Figure 4, the whole JJJ region is treated as a whole. This treatment makes the presentation repetitive, and also makes the logic of this paper quite difficult to follow.**

Response: Accept. Thank you for your valuable comments. Regarding the question (the results are sometimes speculative) you mentioned, we re-examine the applicability of each result in Section 3.4 carefully.

First, we remove some speculative results. For example, the sentences "The CLWP provides partial information about meteorological conditions and cloud types, thus we speculate that the inhomogeneous ACI effect on cloud under haze pollution conditions in Jing-Jin-Ji is related to the magnitude of pre-simulated CLWP, which needs further work to verify." and "which is conducive to cloud droplets nucleation and cloud evolution." in Section 3.4 is deleted.

Second, we add more pictures and references for some speculative results. For example, the sentence "suggesting that the ACI effect is not entirely determined by aerosol levels … " is changed to "This suggests that the ACI effect is probably dominated more by supersaturation and ascent speed, rather than aerosol

concentration, in these subareas of Jing-Jin-Ji. As pointed out by Hudson and Noble (2014), the ACI depends more on ascent speed than aerosol concentration when CCN is larger than 400 cm$^{-3}$ in stratus cloud." in lines 351-354 in manuscript. We also redraw the Figure S7 and give more detailed descriptions of the reasons for changed precipitation in the DA and DB in Text S2 in supplementary material, which is as follows:

Rain tendency= prr_wau + prr_rcw + prr_sml + prr_gml + prr_rcs + prr_rcg - prg_rfz - pri_rfz - prr_rci,

where prr_wau is the autoconversion of cloud water to form rain, prr_rcw is the rain collecting cloud water, prr_sml is the melting of snow to form rain, prr_gml the is melting of graupel to form rain, prr_rcs is the rain collecting snow, prr_rcg is the rain collecting graupel, prg_rfz is the freezing of rainwater into graupel, pri_rfz is the freezing of rainwater into ice, and prr_rci is the rain collecting ice. All of these processes lead to changes in rainwater. Figure S7 shows the difference in mean hydrometeors mixing ratio and rain tendency processes between the E2 (ACI) and E1 (NO-ACI) experiment. In the DA, the largest contribution to the decrease in precipitation is the prr_sml, followed by the prr_rcs and other processes (Figure S7 (c)). In the DB, the largest contribution to the increase in precipitation is also the prr_sml, followed by the prr_rcw and other processes (Figure S7 (d)). In summary, the increase or decrease in simulated precipitation due to the ACI is mainly caused by the melting of snow to form rain (i.e., prr_sml), followed by other source/sink processes of rainwater.

Finally, we have to show that there are indeed great difficulties in determining some results, such as the sentence "Second, if the original model cannot reproduce the observed cloud fields in some areas or periods, the ACI has almost no effect on simulations, which can be likely attributed to the cloud microphysical scheme, the initial fields, etc. (Thompson and Eidhammer, 2014; Fan et al., 2016; White et al., 2017)" in Section 3.4 in lines 365-368. In the current NWP model, the simulation performance for cloud variables is very poor and it is difficult to find a solution. It is expected that new knowledge can be available to improve the model in the future.

[Figure]

**Figure S7:** The difference of mean hydrometeors mixing ratio (top) and rain tendency processes (bottom) between the E2 and E1 experiment on 7 January 2017 in the DA (a and c) and DB (b and d).

As for another question (in Section 3.4, two sub-regions are selected for analysis; while in Figure 4, the whole JJJ region is treated as a whole. This treatment makes the presentation repetitive, and also makes the logic of this paper quite difficult to follow), some reasons are given as follows: 1. As shown in the title (Aerosol cloud interaction in the atmospheric chemistry model GRAPES_Meso5.1/CUACE and its impacts on mesoscale numerical weather prediction under haze pollution conditions in Jing-Jin-Ji in China), the main purpose of this study is to investigate the impact of ACI on the weather forecast in the heavily polluted Jing-Jin-Ji region of China. Thus in Figure 4, we treat the Jing-Jin-Ji region as a whole to analyze the changes in hydrometeors mixing ratio, which facilitates our understanding of how ACI mainly affects the

simulations in Jing-Jin-Ji region and is consistent with the study purpose. 2. After studying the improved performance of the ACI in Jing-Jin-Ji region, we find that even in a localized area, such as Jing-Jin-Ji, the ACI effect varies significantly in time and space, which is described in detail in Sections 3.2 and 3.3 of the manuscript and raises our concerns. We wanted to know the relevant causes of this phenomenon and to deepen our understanding of how the ACI can improve the simulations from the model. 3. Previous studies have reported that the factors affecting the ACI include aerosol concentration, supersaturation, ascent speed, cloud types, overlap degree of cloud and aerosol layers, etc. (Pawlowska and Brenguier, 2000; Lohmann and Feichter, 2005; Zhou et al., 2020; Lu et al., 2012). If the Jing-Jin-Ji region is treated as a whole, the distinct patterns of ACI effect that could possibly exist for different regions within the whole domain will be averaged out, which results in unreasonable analysis. Therefore, a suitable approach is to separate these factors into several subareas, so that each of them could be attributed to a certain aerosol condition and a certain microphysical situation. Using this approach, we select two specific subareas for comparative analysis of the causes affecting the ACI in Section 3.4 based on aerosol concentration and cloud type.

In addition, due to our lack of clarity of statement, it does cause the repetition and logical conflicts between the content in Section 3.4 and the previous sections, which is as you mentioned. To avoid these, we have modified the whole Section 3.4 accordingly in lines 333-369, which is as follows:

[revised manuscript text omitted]

**3. The English language is rough at some places, and needs improvement. Some places are mentioned in the minor comments, but other places are not.**

Response: Accept. We have modified these places you mentioned in the minor comments and improved the English language throughout the manuscript.

Minor comments:

**Line 38: 70% percent is probably near the higher end of global cloud cover reported in the literature, so it might be better to use a reference here.**

Response: Accept. According to previous studies (Ding et al., 2005; Mao et al., 2019), 70% is indeed near the higher end of global cloud cover. We have added the corresponding reference for this sentence "Cloud covers approximately 70% of the Earth's surface (Ding et al., 2005; Mao et al., 2019) …" in line 38.


Response: Revised. The "to be defined" has been changed to "to define" in line 51.

**Figure 1: It is really difficult to see the automatic weather stations from this figure.**

Response: Accept. We have redrawn the Figure 1 to make the automatic weather stations clearer.

[Figure]

**Figure 1: The map and topographic height of the simulated domain. The turquoise line represents a part of the CALIPSO satellite orbit tracks at 18:12 on 7 January 2017, the black rectangle represents the location**

**of Jing-Jin-Ji, the gray cross signs are the automatic weather stations, and the dark red dots are the air pollution stations.**

**Line 142: There are 120 h from 4-8 January, while the prediction time plus the spin up time is 144 h. Is here an error?**

Response: Revised. The descriptions of the prediction time are inaccurate here and may lead to misunderstand, for this we have changed the sentence "The study period is from 4 to 8 January 2017 with 72 h prediction time. The spin-up time is 72 h." to "The whole simulation period is from 30 December 2016 to 10 January 2017 with 72 h as a looping experiment. The results of the first 72 h (30 December 2016 to 1 January 2017) are regarded as the spin-up time to keep the model stable and to avoid the effects of the chemical initial fields. The study period is from 4 to 8 January 2017 (from cloud formation to dissipation in Jing-Jin-Ji) in this paper." in lines 143-146 in the manuscript.

**Line 143: "achievement" to "implementation"?**

Response: Revised. The "achievement" has been amended to "implementation" in line 147.

**Section 2.3. Some important details are not clear or missing here. For example, how droplet activation and ice nucleation are calculated? Which variables in Eq. (1-3) are prognostic by the model and which variables are specified? If specified, what are the values? What are the meaning of tracer number 1-49? Are scavenging processes considered? And some other details. I understand that some details might be included in the references, but a concise description might be helpful for the readers.**

Response: Accept. Thanks to your suggestions, some important details of the changes we make in the model have been added to the supplementary material in Text S1 and the manuscript in lines 150-169.

(1) The cloud droplets activation

In the new Thompson cloud microphysics scheme, when the supersaturation degree is greater than 0, the water-friendly aerosol can be activated as cloud droplets by equation (1):

$$Activ\_Nc= NWFA \times AF \qquad (1),$$

where Activ_Nc represents activated cloud droplets, NWFA represents the number concentration of water-friendly aerosol, and AF represents activated fraction. The activation fraction is determined by the simulated ambient temperature (K), vertical

velocity (m/s), number concentration of water-friendly aerosol (cm$^{-3}$), and pre-determined values of the hygroscopicity parameter (0.4) and aerosol mean radius (0.04 μm) by using a lookup table. This lookup table is created by the explicit treatment of Köhler activation theory using different number concentration of water-friendly aerosol (10.0, 31.6, 100.0, 316.0, 1000.0, 3160.0, 10000.0 cm$^{-3}$), vertical velocity (0.01, 0.0316, 0.1, 0.316, 1.0, 3.16, 10.0, 31.6, 100.0 m/s), temperature (243.15, 253.15, 263.15, 273.15, 283.15, 293.15, 303.15 K), aerosol mean radius (0.01, 0.02, 0.04, 0.08, 0.16 μm), and hygroscopicity parameter (0.2, 0.4, 0.6, 0.8) according to previous studies (Feingold, and Heymsfield, 1992; Thompson and Eidhammer, 2014).


Water-friendly aerosol number concentration (/kg) required by the activation in the cloud microphysics scheme is calculated by aerosol mass concentration at each grid point according to equations (3), (4), and (5):

$$m_{num} = \frac{4}{3} \times \pi \times r_{num}^3 \times (\rho_{num}) \qquad (3),$$

$$N(i, k, j, num) = tracer(i, k, j, num)/m_{num} \qquad (4),$$

$$NWFA2(i, k, j) = \sum_{num=1}^{49} N(i, k, j, num) \qquad (5).$$

Here, the m is the aerosol mass (kg), the num is the tracer number from 1 to 49, the r is the mean radius (μm), the ρ is the aerosol density (g cm$^{-3}$), the tracer is the aerosol mass concentration (kg/kg), the N is the aerosol number concentration (/kg), and the NWFA2 is the total water-friendly aerosol number concentration (/kg). I, j, and k represent the grid point. The tracer is the prognostic variable. The num, r, and ρ are specified in Table S1.

Table S1: The specified values of the tracer number, aerosol types, mean radius (r), and density (ρ).

| Tracer number | Aerosol types | Mean radius (μm) | Density (g cm$^{-3}$) |
|---|---|---|---|
| 1 | OC1 | 0.0075 | 1.30 |
| 2 | OC2 | 0.015 | 1.30 |
| 3 | OC3 | 0.03 | 1.30 |
| 4 | OC4 | 0.06 | 1.30 |
| 5 | OC5 | 0.12 | 1.30 |
| 6 | OC6 | 0.24 | 1.30 |
| 7 | OC7 | 0.48 | 1.30 |
| 8 | OC8 | 0.96 | 1.30 |
| 9 | OC9 | 1.92 | 1.30 |
| 10 | OC10 | 3.84 | 1.30 |
| 11 | OC11 | 7.68 | 1.30 |
| 12 | OC12 | 15.36 | 1.30 |
| 13 | SS1 | 0.0075 | 2.17 |
| 14 | SS2 | 0.015 | 2.17 |
| 15 | SS3 | 0.03 | 2.17 |
| 16 | SS4 | 0.06 | 2.17 |
| 17 | SS5 | 0.12 | 2.17 |
| 18 | SS6 | 0.24 | 2.17 |
| 19 | SS7 | 0.48 | 2.17 |
| 20 | SS8 | 0.96 | 2.17 |
| 21 | SS9 | 1.92 | 2.17 |
| 22 | SS10 | 3.84 | 2.17 |
| 23 | SS11 | 7.68 | 2.17 |
| 24 | SS12 | 15.36 | 2.17 |
| 25 | SF1 | 0.0075 | 1.79 |
| 26 | SF2 | 0.015 | 1.79 |
| 27 | SF3 | 0.03 | 1.79 |
| 28 | SF4 | 0.06 | 1.79 |

| 29 | SF5  | 0.12   | 1.79 |
|----|------|--------|------|
| 30 | SF6  | 0.24   | 1.79 |
| 31 | SF7  | 0.48   | 1.79 |
| 32 | SF8  | 0.96   | 1.79 |
| 33 | SF9  | 1.92   | 1.79 |
| 34 | SF10 | 3.84   | 1.79 |
| 35 | SF11 | 7.68   | 1.79 |
| 36 | SF12 | 15.36  | 1.79 |
| 37 | NT1  | 0.0075 | 1.77 |
| 38 | NT2  | 0.015  | 1.77 |
| 39 | NT3  | 0.03   | 1.77 |
| 40 | NT4  | 0.06   | 1.77 |
| 41 | NT5  | 0.12   | 1.77 |
| 42 | NT6  | 0.24   | 1.77 |
| 43 | NT7  | 0.48   | 1.77 |
| 44 | NT8  | 0.96   | 1.77 |
| 45 | NT9  | 1.92   | 1.77 |
| 46 | NT10 | 3.84   | 1.77 |
| 47 | NT11 | 7.68   | 1.77 |
| 48 | NT12 | 15.36  | 1.77 |
| 49 | AM   | 0.06   | 1.69 |

(4) The wet scavenging of aerosol and evaporation of cloud droplets

The wet scavenging of aerosol can be divided into the in-cloud and below-cloud scavenging. We calculate the in-cloud scavenging process of aerosol by the collision-coalescence process between aerosol and raindrops (Giorgi and Chameides, 1986). The evaporation of raindrops will lead to returned aerosol. In the below-cloud scavenging process, the removal of aerosol is calculated by using the rain/snow scavenging rate according to previous studies (Slinn, 1977; Gong et al, 1997). All of these wet scavenging processes are activated in the CUACE aerosol module and can give real-time feedback to the aerosol field. The activation of aerosol as cloud droplets does not update the aerosol field in the current model version. In the future, we will complete this process in the GRAPES_Meso5.1/CUACE model and study the impact of ACI on aerosol in detail.


**Figure 3. Specify which experiment is presented here. More importantly, why do the authors display the whole simulated domain instead of JJJ only, since almost all discussions focus on JJJ?**

Response: Revised. In Figure 3, the left column represents the observations and the right column represents the simulations from the E1 experiment without the ACI, which has been specified in Figure 3 and the Figure legend in line 220.

We understand the doubts about why we display the whole simulated domain instead of Jing-Jin-Ji only, even though almost all discussions focus on Jing-Jin-Ji. To this end, we have the following reasons: 1. Improve the understanding of the simulation performance in Jing-Jin-Ji. Aerosol pollution and meteorological conditions in Jing-Jin-Ji are closely linked to those in other regions, such as the transboundary transport of haze between the Yangtze River Delta and the Jing-Jin-Ji (Huang et al., 2020). A similar phenomenon is observed in meteorological conditions. 2. More clearly identify the case of selected cloud systems. Small-scale cloud has large uncertainties and is not easily simulated by the numerical weather prediction model, which will make it difficult to study the ACI; while in this paper, it can be seen from Figure 3 that the selected case is the large-scale cloud system, which is more easily simulated by the model. 3. Demonstrate the overall performance of the model. Figure 3 shows that this model has good performance for aerosol pollution and meteorological conditions in the whole simulated domain, not only in Jing-Jin-Ji. This suggests that the ACI studies can potentially be carried out in other regions in the future.

**Line 204: "reasonably" to "reasonable"?**

Response: Corrected. The "reasonably" is changed to "reasonable" in line 208.

**Figure 4. Why is the spin-up period also displayed?**

Response: We understand the doubts about the spin-up period. Actually, the spin-up period is from 30 December 2016 to 1 January 2017, which has been discarded to avoid the effects of the chemical initial fields.

**Line 275: It may be inappropriate to present the maximum value of decrease, because it may not be representative. For example, a slight shift of cloud position may cause a large change in SDSR.**

Response: Accept. We have deleted the presentation of the maximum value of decreased SDSR in the revised manuscript.

**Figure 9 and related discussions: What are the criterion of selecting the light-rain region and the moderate-rain region? In the northeast of the JJJ-region, there is also a contiguous precipitation region, which is this region not analyzed?**

Response: Revised. We select the moderate and light rainfall regions to study separately because the response of different types of precipitation to changed aerosol

concentration is different. In this paper, a rainfall event is selected from an automatic weather station within 24 h cumulative precipitation>0 mm. We define a moderate (light) rainfall event as 10 mm<24 h cumulative precipitation<25 mm (0.1 mm<24 h cumulative precipitation < 10 mm). If all rainfall events from contiguous stations in a certain region are moderate rainfall, this region is defined as the moderate rainfall region. Similar procedures are applied to the light rainfall region. The criterion of selecting the light rainfall region and the moderate rainfall region is added to the manuscript in lines 311-316.

As you mentioned, there is also a contiguous precipitation region in the northeast of the Jing-Jin-Ji region (the region ② in Figure R1(a)), which can be defined as the light rainfall region. We have investigated the impact of ACI on 24 h cumulative precipitation in this light rainfall region ② and compared it with the region ①. We find that the ACI has little effect on the precipitation in the light rainfall region ② with the maximum absolute value of changes less than 0.2 mm (Figure R1(b)); while the impact of ACI on the precipitation in the light rainfall region ① is more significant with the maximum value of decrease exceeding -1 mm (Figure R1(b)). The analysis of the simulated cloud in the E1 and E2 experiment also reveals that the simulations cannot reproduce the location of cloud fields from the VIIRS over the region ② (Figure R1(c)), and the ACI cannot improve this phenomenon (Figure R1(d)). To reflect the impact of ACI in the light rainfall region, we only select the region ①, rather than the region ②, as an example to represent the light rain region in the second paragraph of Section 3.3.

[Figure]

**Figure R1: The spatial distribution of 24 h cumulative precipitation and CLWP on 7 January 2017. (a) The observations. (b) The difference between the E2 and E1 experiment. (c) The VIIRS. (d) The difference between the E2 and E1 experiment. The red and black ovals represent the moderate and light rainfall regions, respectively.**

**Line 333-335: Please clarify whether the absolute value or the relative values are compared. For example, the authors used "more significant" in line 334, however, the percentage change is "less significant".**

Response: Revised. Based on your suggestions, we have clarified that the absolute values of changed CLWP and COT are compared and thus the more significant increase of CLWP and COT occurs in the DB. The corresponding changes have been added to the manuscript in lines 347-348.

**Line 339: Aerosol level might also refer to aerosol height. Change "level" to "concentration" might be better.**

Response: Accept. We have changed the "aerosol levels" to "aerosol concentration" throughout the manuscript.

**Line 343: A brief introduction of how supersaturation is calculated is helpful in interpreting the results here.**

Response: Accept. In the Thompson cloud microphysics scheme, the supersaturation degree (S) is obtained by the following function (6):

$$S = \left(\frac{q}{q_s} - 1\right) \times 100\% \tag{6},$$

where q and $q_s$ represent the water vapor mixing ratio and the saturation water vapor mixing ratio. The $q_s$ is given by the function (7) and (8):

$$q_s = \frac{0.622 \times e_{sw}}{p - e_{sw}} \qquad (7),$$

$$e_{sw} = 6.112 \times \exp\left\{17.67 \times \left(\frac{T - 273.16}{T - 29.65}\right)\right\} \qquad (8),$$

where $e_{sw}$ is the saturated vapour pressure over water (hPa), p is the air pressure (hPa), and T is the air temperature (K). All of these have been added to the manuscript in lines 150-151 and to the supplementary material in Text S1.1.

**Line 345: What is "pre-calculated"?**

Response: In the original manuscript, we want to use the "pre-simulated (i.e., pre-calculated)" to represent the results simulated by the original scheme without ACI. To facilitate understanding, we have deleted the "pre-simulated (i.e., pre-calculated)" in the revised manuscript.